

# Predicting migration routes for three species of migratory bats using species distribution models

Jamin G. Wieringa[1,2], Bryan C. Carstens[1] and H Lisle Gibbs[1,2]

[1] Department of Evolution, Ecology and Organismal Biology, The Ohio State University, Columbus, OH, USA
[2] Ohio Biodiversity Conservation Partnership, The Ohio State University, Columbus, OH, USA

## ABSTRACT

Understanding seasonal variation in the distribution and movement patterns of migratory species is essential to monitoring and conservation efforts. While there are many species of migratory bats in North America, little is known about their seasonal movements. In terms of conservation, this is important because the bat fatalities from wind energy turbines are significant and may fluctuate seasonally. Here we describe seasonally resolved distributions for the three species that are most impacted by wind farms (*Lasiurus borealis* (eastern red bat), *L. cinereus* (hoary bat) and *Lasionycteris noctivagans* (silver-haired bat)) and use these distributions to infer their most likely migratory pathways. To accomplish this, we collected 2,880 occurrence points from the Global Biodiversity Information Facility over five decades in North America to model species distributions on a seasonal basis and used an ensemble approach for modeling distributions. This dataset included 1,129 data points for *L. borealis*, 917 for *L. cinereus* and 834 for *L. noctivagans*. The results suggest that all three species exhibit variation in distributions from north to south depending on season, with each species showing potential migratory pathways during the fall migration that follow linear features. Finally, we describe proposed migratory pathways for these three species that can be used to identify stop-over sites, assess small-scale migration and highlight areas that should be prioritized for actions to reduce the effects of wind farm mortality.

## INTRODUCTION

Conservation and management of migratory animals requires knowledge about their seasonal movements across space and time (*Webster et al., 2002*). In a wide variety of taxa, species migrate when resources vary seasonally (*Shaw & Couzin, 2013*) or temperature variation results in thermal stress (*Fleming & Eby, 2003*). Due to small body sizes it is difficult to track long distance movements of many taxa such as species of bats, birds and insects reducing our understanding of their migratory behavior. While some progress has been made using light-level geolocators (*Åkesson et al., 2012*) and various biomarkers (e.g., *Hobson & Wassenaar, 2018*), these methods have limitations such as requiring

Corresponding author
Jamin G. Wieringa,
wieringa.3@osu.edu

recapture and low precision, respectively and as a result are limited in their impact. This is particularly true for bats, small-bodied nocturnal mammals capable of true flight.

Although many species of bats migrate, only 12 of 500 Vespertilionid bats undertake long-distance migration and understanding their migration is vital to the conservation of these species (*Fleming & Eby, 2003*; *Simmons & Cirranello, 2020*; *Welbergen et al., 2020*). By understanding the migration of these species, we can better understand the pressures an individual will face during migration or at home ranges during non-migratory time periods. However, limited information is currently available about the long-distance migration of bats in North America. For example, in most species the approximate direction (north-south), time of year and some rough estimates of distances travelled are all that is known (*Fleming & Eby, 2003*; *Pettit & O'Keefe, 2017*) with much of this information inferred from distribution modeling in these species (e.g., *Hayes, Cryan & Wunder, 2015*) or biomarker studies, such as isotopes (e.g., *Cryan, Stricker & Wunder, 2014*).

Hydrogen isotopes have largely been the focal method for investigations into migration of North American tree bats. For example, *Baerwald, Patterson & Barclay (2014)* used isotope information to propose that *Lasiurus cinereus* and *Lasionycteris noctivagans* use the eastern slopes of the Rocky Mountains as a migration route. Further, *Cryan, Stricker & Wunder (2014)* used stable isotope data to suggest that *L. cinereus* has some east-west movement during migration in addition to the north-south, likely toward coastal regions during Autumn migration that potentially contain more suitable winter habitat.

In contrast to our understanding of migration in North American bats, there is more known about bats from Europe and other regions. Previous studies have shown repeated and partial migration (*Lehnert et al., 2018*), and that bats showed site fidelity at stop-over sites during migration (*Giavi et al., 2014*). One important aspect of research is understanding how bats navigate during migration and some have suggested the tracking of linear features for bat migration (e.g., *Voigt et al., 2016*; *Ahlén, Baagøe & Bach, 2009*), although others have challenged this interpretation (*Krauel, McGuire & Boyles, 2018*). Further studies have also shown the impacts of humans during migration. Human activities have the potential to disrupt bat migration via mechanisms such as interfering with magnetic navigation (*Voigt et al., 2017*), increasing light pollution (*Lacoeuilhe et al., 2014*), developing wind farms along migration corridors (*Hayes, Cryan & Wunder, 2015*), or reducing stop-over sites and food availability through deforestation and habitat destruction. To mitigate these effects, a better understanding of migration in bats is needed.

A few of these species such as *L. borealis* (Eastern Red bat), *L. cinereus* (Hoary bat) and *L. noctivagans* (Silver-haired bat) have been a focus of understanding these behaviors in North America due to their high mortality at wind farms, with some estimates predicting a 90% species population decline within 50 years due to wind farm interactions (*Frick et al., 2017*). Bats are the most common animal found dead beneath wind turbines in North America (*Kunz et al., 2007*) with the majority (~80%) of these individuals consisting of just the three species of migratory listed above (*Arnett & Baerwald, 2013*). Most of the fatalities for these species occur during a period of time coinciding with

autumn migration (*Kunz et al., 2007*), but data linking the act of migration and mortality is lacking. Overall, a more precise delineation of possible migratory corridors (defined here as the most likely path followed during migration) and how these influence wind farm interactions could help to minimize impacts of wind facilities on these species.

One explanation for the uncertainty about long-distance migratory pathways of migratory bats is the lack of data on spatial locations through time which is in contrast to such data which are widely available in better-studied migratory species such as birds. One reason for this difference is that observational data on birds can come from a variety of Citizen Science initiatives such as the Breeding Bird and Christmas Bird Surveys and eBird (*National Audubon Society, 2010*; *Sullivan et al., 2009*). The difference in data quantity is large. For example, a common migratory bird, the yellow warbler (*Setophaga petechia*) has 2.39 million occurrences the Global Biodiversity Information Facility (GBIF; checked 11 December 2019) whereas the entire family of Vespertilionidae bats consisting of >400 species have only 1.49 million occurrences recorded. While the causes of this disparity are many, the difference highlights the need to use other sources of data for to study broad scale patterns of bat migration.

One approach to better understand seasonal distributions and identify migratory corridors is to generate seasonally explicit species distribution models (SDMs; *Fink et al., 2010*; *Hayes, Cryan & Wunder, 2015*; *Smeraldo et al., 2018*) and use these to infer movement patterns. This approach has been successful in other migratory species, such as birds (*Reynolds et al., 2017*). While other studies have begun to explore this approach with tree bats (see *Findley & Jones, 1964*; *Cryan, 2003*; *Hayes, Cryan & Wunder, 2015*), much of this work has focused on overall distributions as opposed to seasonal differences in movement. However, seasonal movements can be studied by generating models of bat distributions on a month-by-month basis that allow seasonal variation to be visualized and infer the movement that took place between monthly occurrences.

In this study, our objective was to identify possible migratory pathways utilized by migratory bat species in North America (*L. borealis*, *L. cinereus* and *L. noctivagans*) through modelling their seasonal distributions using SDMs. While not the first study to generate SDMs for these species, it is the first to use them to infer migration patterns for the time between seasonal occurrences. The models generated in this study shed light on the seasonal dynamics for these three species and highlight areas of interest for further study of migratory corridors that could be used to investigate stop-over sites, small scale migration, and be used as a starting point for designing methods to mitigate wind farm mortality.

# METHODS

## Occurrence data from GBIF

Figure 1 shows an overview of the steps involved in data collection and analysis; more detailed methods are described on Supplemental Material. To begin, all available occurrence data were downloaded for *L. borealis*, *L. cinereus*, and *L. noctivagans* through the GBIF data portal (http://www.gbif.org) on 11 March 2019 using only 'Preserved Specimens,' 'Human Observations' and 'Material Sample' keywords for data from the past
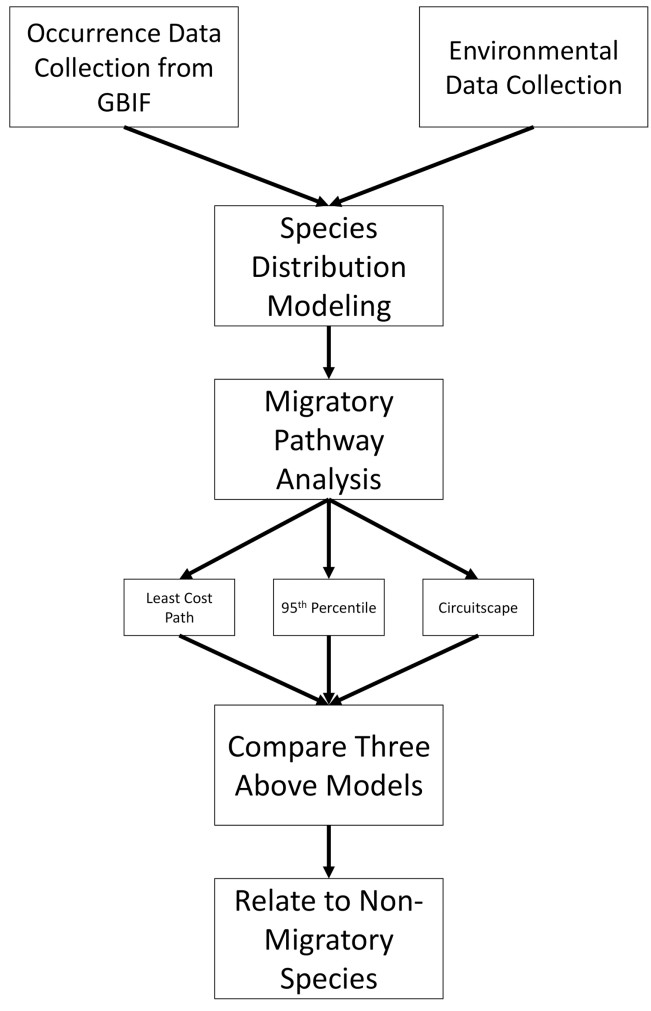

**Figure 1 Methods overview.** Flowchart showing how occurrence data were analyzed and used to infer migratory pathways for each bat species.  

50 years (https://doi.org/10.15468/dl.dpiwzi, https://doi.org/10.15468/dl.irfol0 and https://doi.org/10.15468/dl.viiyt5, respectively). This 50-year period was selected because it allows for more confidence in the call of a species and its locality. While some previous studies have verified occurrence data from older than 50 years ago (see *Hayes, Cryan & Wunder, 2015*), we were unable to access the data from the museum collections used in that study. In addition, due to climate change ranges may be shifting as has been observed in some mammal species (*MacLeod, 2009*) and so using more recent occurrence data likely provides more accurate current ranges. All downloaded records were screened using several filters (see Supplemental Methods) as recommended by others (*Feeley & Silman, 2011*; *Carstens et al., 2018*). Once data sets were filtered using these criteria, we corrected for over sampling within a 1° region following guidelines given by *Hijmans & Elith (2017)*. In brief, we created a grid of 1° resolution (~111 km) and subsampled our occurrence data to one occurrence per grid cell. This was done to reduce the possibility of sampling bias in our data. While there exists a possibility of overthinning

data, a larger concern was sampling bias inflating suitability of certain regions especially on these analyses as suitability will determine the inferred pathways. This is especially true for the known spatial bias that exists in GBIF data, due to differences in funding and data sharing of institutions (*Razgour et al., 2016*). As a result, we took the conservative approach of 1° grid sampling. In addition, other continent-wide studies for some species have used similar scales for filtering and were shown to be an effective filtering approach (*Fourcade et al., 2014*). Lastly, some have suggested that sampling be limited to an approximate home range for a species (*Kramer-Schadt et al., 2013*). In this case, due to the highly volant nature of bats with the ability to travel long distances of a given night broader filtering is needed. For example, *Morningstar & Sandilands (2019)* documented an individual *L. cinereus* traveling over 800 km in two weeks, finishing ~50 km from the initial roost. As a result of this high mobility, a larger grid sample likely produces more accurate distribution models.

## Predictor environmental variables

WorldClim version 2 monthly climatic data were used at 2.5-min resolution (~4.5 km; *Fick & Hijmans, 2017*) for our species distribution models and included the following variables: precipitation (mm), solar radiation (kJ m$^{-2}$ day$^{-1}$), average temperature (°C), maximum temperature (°C), minimum temperature (°C), vapor pressure (kPa) and wind speed (m s$^{-1}$; downloaded on 3 June 2019 from worldclim.org). WorldClim is a database of high spatial resolution global weather and climate data. These data can be used for mapping and spatial modeling. Additionally, elevation maps (*Tachikawa et al., 2011*; 11 March 2019), and the human influence index (*WCS & CIESIN, 2005*; 11 March 2019) for North America were also downloaded as *Jung & Threlfall (2018)* showed a negative response to urbanization in the Americas in insectivorous bats in the family Vespertilionidae. Human influence was determined by combining population density, human land use and infrastructure, and human access (*WCS & CIESIN, 2005*). Following Hayes et al. (2015), we also included MODIS Normalized Difference Vegetative Index (*Didan et al., 2015*) and Global Tree Coverage 2010 (*Hansen et al., 2013*) as metrics of seasonality and leaf growth, which could impact prey abundance, and be a metric of available roost sites in trees, downloaded on 4 June 2019 and 5 June 2019, respectively. Prior to final selection of predictor variables, correlations between each possible pair of predictor variables were determined and one variable from each pair that was strongly correlated with the other was removed ($r > 0.8$; *Mateo et al., 2013*). Any removal of a variable was determined based on biological relevance and previous uses in literature.

## Species distribution modeling

Species distribution models were generated for each species using five different methods: four specific model algorithms and an ensemble approach (see below). Specific algorithms included: generalized linear model (GLM; *Guisan, Edwards & Hastie, 2002*), BIOCLIM model (BC; *Booth et al., 2014*), random forest (RF; *Breiman, 2001*; *Mi et al., 2017*) and maximum entropy (MaxEnt; *Phillips, Dudik & Schapire, 2017*). These four approaches, while good predictors individually, can be made more effective through an ensemble method. This approach accounts for the problems of each model and can allow for better
performing models (*Araújo & New, 2007*; *Marmion et al., 2009*) and is becoming more common (*Razgour et al., 2016*). Due to this and our results (see below), we used the ensemble models for all analyses.

All SDM analyses were carried out in R using the packages "randomForest" (*Liaw & Wiener, 2018*), "raster" (*Hijmans, 2019*), "rgeos" (*Bivand & Lewin-Koh, 2019*), "maptools" (*Bivand & Lewin-Koh, 2019*), "dismo" (*Hijmans et al., 2017*), "sp" (*Pebesma & Bivand, 2012*), "ecospat" (*Di Cola et al., 2017*), "maps" (*Becker et al. 2018*), and "rJava" (*Urbanek, 2019*). We created 1,000 pseudo absence points for each month from random points in the background layers and partitioned the model into testing (80%) and training data (20%) using the "kfold" function.

Each model was then assessed using the Area under the receiver operating characteristic curve (AUC) and the True Skill Statistic (TSS). These values were then used to weigh each layer and were then combined into a single ensemble SDM. Following generation of our ensemble models, they were assessed using the same AUC and TSS metrics as outlined above and data points used for all other models for comparison to determine which model to use for further analysis. These layers were then used to predict migratory pathways.

The importance of individual variables was assessed using different methods for each model. For RF we used the 'importance' function in the "randomForest" R package to measure the importance of a variable in a model. With MaxEnt, variable importance was assessed using 'var.importance' function in "ENMeval" to determine the importance of each variable (*Muscarella et al., 2014*). For the GLM model, we used the 'varImp' function present in "caret" (*Kuhn et al., 2020*).

## Migratory pathways

To identify migratory pathways using SDMs we used three complementary methods: circuit theory (*McRae & Beier, 2007*; *Shah & McRae, 2008*), 95th percentile suitability (*Poor et al., 2012*), and least cost path analyses (LCP; *Howey, 2011*). Since each of these methods have advantages and disadvantages, results from these three methods were compared to generate a consensus delineation of possible corridors (*Bond et al., 2017*; *Marrotte & Bowman, 2017*). While some authors have argued for selecting the single best hypothesized approach (*Marrotte & Bowman, 2017*), as we do not know if these species follow linear features as has been observed in some species (*Ahlén, Baagøe & Bach, 2009*) or exhibit more erratic movements, we could not confidently select a single approach. Multiple authors (*Dutta et al., 2016*; *Medley, Jenkins & Hoffman, 2015*) show that circuit and least-cost based analyses complement each other and can give more insight into the movement of a species. In addition, the use of circuit theory and least-cost-path allow for some movement through less suitable areas. In least-cost-path analyses an individual's path may go through less suitable areas as the model weighs both distance and suitability of the areas traveled through. For example, the model would give preference for travel through a single less suitable cell if the cumulative cost is less than four more suitable cells. Similarly, in circuit theory it is possible for the "current" (an individual
movement likelihood) to pass through less suitable cells as long as the total resistance to do so is less than adjacent alternative paths.

For circuit theory, the protocol of *Burke et al. (2019)* was followed. In brief, we aggregated our winter month occurrences (December–February) into a single dataset and did the same for summer months (June–July), using Hayes et al. (2015) to determine the appropriate months for each season. As SDMs can be interpreted as conductance maps, we used an average of both spring and fall months (March, April, May and August, September, October, respectively) to assess potential corridors between winter and summer occurrences. These time periods are based on previously published distributions of occurrences (*Cryan, 2003*), previous SDM modeling (*Hayes, Cryan & Wunder, 2015*), wind farm fatality data (*Arnett et al., 2008*), radio telemetry (*Walters et al., 2006*), and acoustic data (*Muthersbaugh et al., 2019*). Using Circuitscape (*Shah & McRae, 2008*), we set our start ("source") and end ("ground") points based on the hypothesized direction of migration. To identify patterns of spring migration, we set our start as winter occurrences and end as summer points, with the spring SDMs as the conductance raster; and summer as start and winter as end with fall SDMs as the fall migration conductance raster; this was repeated for each species.

To use least cost path analysis to predict migratory pathways we used the R function 'shortestPath' implemented in 'gdistance' (*Van Etten, 2017*). The analysis was done iteratively between all points previously designated as "Winter" and "Summer" points for Circuitscape, and spring/fall conductance surfaces for cost determination. As single pathways are likely not informative for species-wide migratory pathways, we combined each least cost path to create a density of pathways. A high density of overlapping paths was used to identify a potential migratory pathway. Additionally, while we are unable to infer if a proposed path is true, we used Moran's I (*Moran, 1950*) and Geary's C (*Geary, 1954*) to quantify if these proposed pathways are positively clustered, as would be expected in a migratory corridor. We also quantified the distance traveled compared to straight-line distance to determine if the proposed pathways would be biologically relevant (i.e., if not following straight line, other factors influence where bats migrate through). Next, binary rasters identifying potential migratory pathways using the 95th percentile approach was generated to identify areas where bats are more likely to be concentrated compared to background (*Poor et al., 2012*). This was to identify areas where suitability is higher and therefore a potential migratory pathway. Finally, overlaps between Circuitscape, least cost path and 95th percentile approaches were identified to highlight locations where they agreed and those were inferred to be potential migratory pathways.

To ensure we are tracking migration and not simply sampling bias, a comparison between the results for migratory pathways above and those from two non-long-distance migratory species (*Myotis lucifugus* and *Eptesicus fuscus*) following the same methods above was carried out. If the pathways are similar to those from these two species it is possible that we are tracking the ability to capture bats during the winter instead of actual movement. On the other hand, if pathways are different, then it is more likely that we are identifying true pathways. Occurrence data for these additional species were collected from GBIF on 31 January 2020 (https://doi.org/10.15468/dl.fphagx) and were treated in
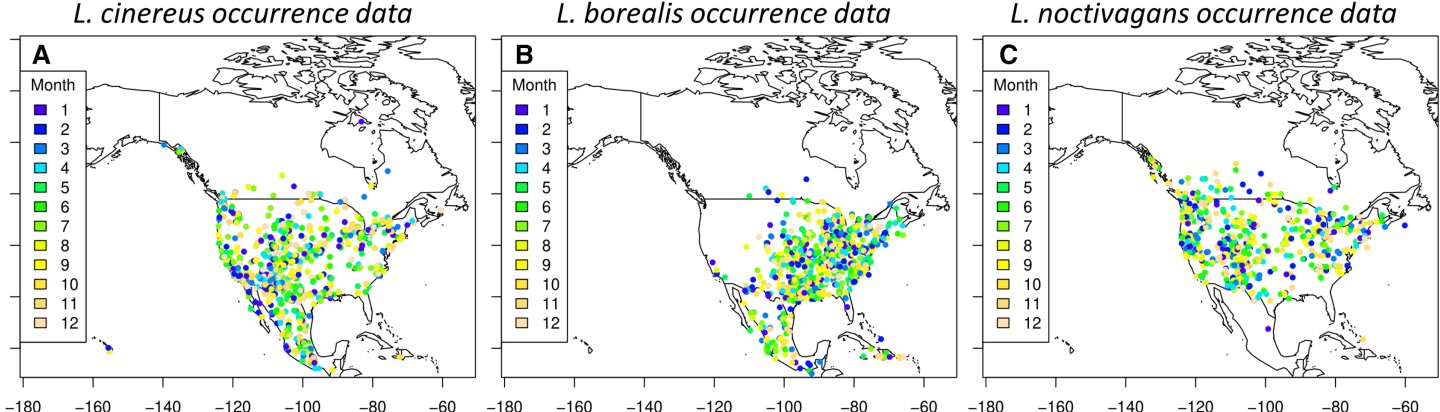

**Figure 2 Occurrences used for species distribution models.** Each dot indicates a filtered occurrence for a migratory species. Each of these points are then given a color based on the month when the occurrence was detected. These points were then used in order to generate species distribution models for each season of a species. (A) Occurrence data for *L. cinereus*, (B) Occurrence data for *L. borealis* and (C) occurrence data for *L. noctivagans*. Map outline generated using the 'maps' R package (*Becker et al., 2018*).

**Table 1 GBIF occurrence data.**

|  | January | February | March | April | May | June | July | August | September | October | November | December | Total |
|---|---|---|---|---|---|---|---|---|---|---|---|---|---|
| *L. borealis* | 45 | 33 | 44 | 75 | 99 | 151 | 205 | 192 | 123 | 86 | 49 | 27 | 1129 |
| *L. cinereus* | 29 | 38 | 44 | 80 | 99 | 108 | 125 | 138 | 110 | 86 | 36 | 24 | 917 |
| *L. noctivagans* | 25 | 24 | 28 | 62 | 110 | 111 | 104 | 131 | 106 | 74 | 38 | 21 | 834 |
| Total | 99 | 95 | 116 | 217 | 308 | 370 | 434 | 461 | 339 | 246 | 123 | 72 | 2880 |

**Note:**
Number of GBIF occurrence points per month for each species analyzed after filtering.

the same manner as the three migratory species to generate SDMs and test migratory pathways.

# RESULTS

## GBIF occurrence data

A total of 20,697 occurrences were downloaded from the GBIF database: 8,362 for *L. borealis*, 7,649 for *L. cinereus*, and 4,686 for *L. noctivagans*. After filtering, there were 10,743 data points remaining: 4,380 for *L. borealis*, 3,736 for *L. cinereus* and 2,627 for *L. noctivagans*. Finally, after accounting for sampling bias there were 1,129 data points for *L. borealis*, 917 for *L. cinereus*, and 834 for *L. noctivagans* (Fig. 2). For each month numbers of data points ranged between 21 and 205 (Table 1). All months were above the minimum of 13 observations suggested by *Proosdij et al. (2016)* (based on simulated data) as necessary for SDM analyses for wide ranging species. Further, only December for two species had occurrences below a higher secondary minimum threshold of 25 data points based on empirical data (per *Proosdij et al., 2016*). SDM analyses were conducted with each subset of data using each of the five modeling approaches: GLM, BC, RF, MaxEnt and ensemble, for a total of 60 model runs for each species. While we acknowledge the presence of other datasets (see NABat (https://www.nabatmonitoring.org/) and/or American Wind/Wildlife Institute (https://awwi.org/)), we found that we had sufficient

**Table 2 Variable importance.**

| | Precipitation | Solar radiation | Temperature | Vapor pressure | Wind speed | Human influence | Elevation | NDVI | Forest cover |
|---|---|---|---|---|---|---|---|---|---|
| Random forest | 5 | 2 | 1 | 3 | 8 | 7 | 4 | 6 | 9 |
| MaxENT | 6 | 2 | 1 | 3 | 8 | 4 | 5 | 7 | 9 |
| GLM | 6 | 1 | 2 | 3 | 7 | 4 | 5 | 8 | 9 |

**Note:**
Variable importance rank for three of the four SDM models implemented in these analyses. 1—indicates the most important variable, while 9—represents the least important. Each importance was found by the following: RF we used the 'importance' function in the "randomForest" R package, MaxEnt, variable importance was assessed using 'var.importance' function in "ENMeval", GLM model, we used the 'varImp' function present in "caret".

data available via GBIF for all months given that we had over 25 occurrences for 11 months, and the only month below this threshold is also deficient in other datasets. Further these other datasets are not comprehensive across North American for all months and/or are not readily available to the public.

## Predictor variables

Following removal of variables that were highly correlated ($r > 0.8$), eight variables were retained: elevation, forest coverage, NDVI, precipitation, solar radiation, average temperature, vapor pressure and wind speed. The variables that were removed were minimum and maximum temperature which were highly correlated with average temperature ($r = 0.98$ and $0.99$, respectively). Average temperature was selected due to the ability of bats to regulate their body temperature and energy expenditure through torpor (*Baloun & Guglielmo, 2019*). While relative importance of variables fluctuated between the four original models implemented (GLM, MaxEnt, RF and BC), in general, average temperature, solar radiation and vapor pressure were the most important variables (Table 2; specific weights Table S1). In contrast, NDVI, percent forest, wind and precipitation were consistently the least important variables.

## Species distribution models

AUC scores range from 0.50 to 0.99, while TSS values range from 0.44 to 0.95 across all five types of models. When evaluated by both AUC and TSS, the consistently best performing species distribution model was the TSS weighted ensemble model (Fig. S1), with this model having a minimum AUC of 0.94 and TSS of 0.78 (Table S2), indicating a high predictive performance (e.g. *Smeraldo et al., 2018*). In addition, these high values indicate sufficient sampling (both number and spatial scale) present for the analyses completed. With the exception of the model for *L. borealis* for July, our TSS weighted ensemble model was always determined to be the best model by both AUC and TSS. As a result of the ensemble models consistently high performance, it was used for all subsequent analyses. We now describe the results for each of the three species.

While we did not explicitly explore the seasonal variation present in each species generated SDM's, this variation can be observed in Fig. 3 (each species detailed in Figs. S2–S4). In short, we observe trends that are similar to those found in previous studies (e.g. *Baerwald & Barclay, 2011*; *Johnson et al., 2011*; *Hayes, Cryan & Wunder, 2015*). For *L. borealis* and *L. cinereus* we observe concentration of habitat suitability in the

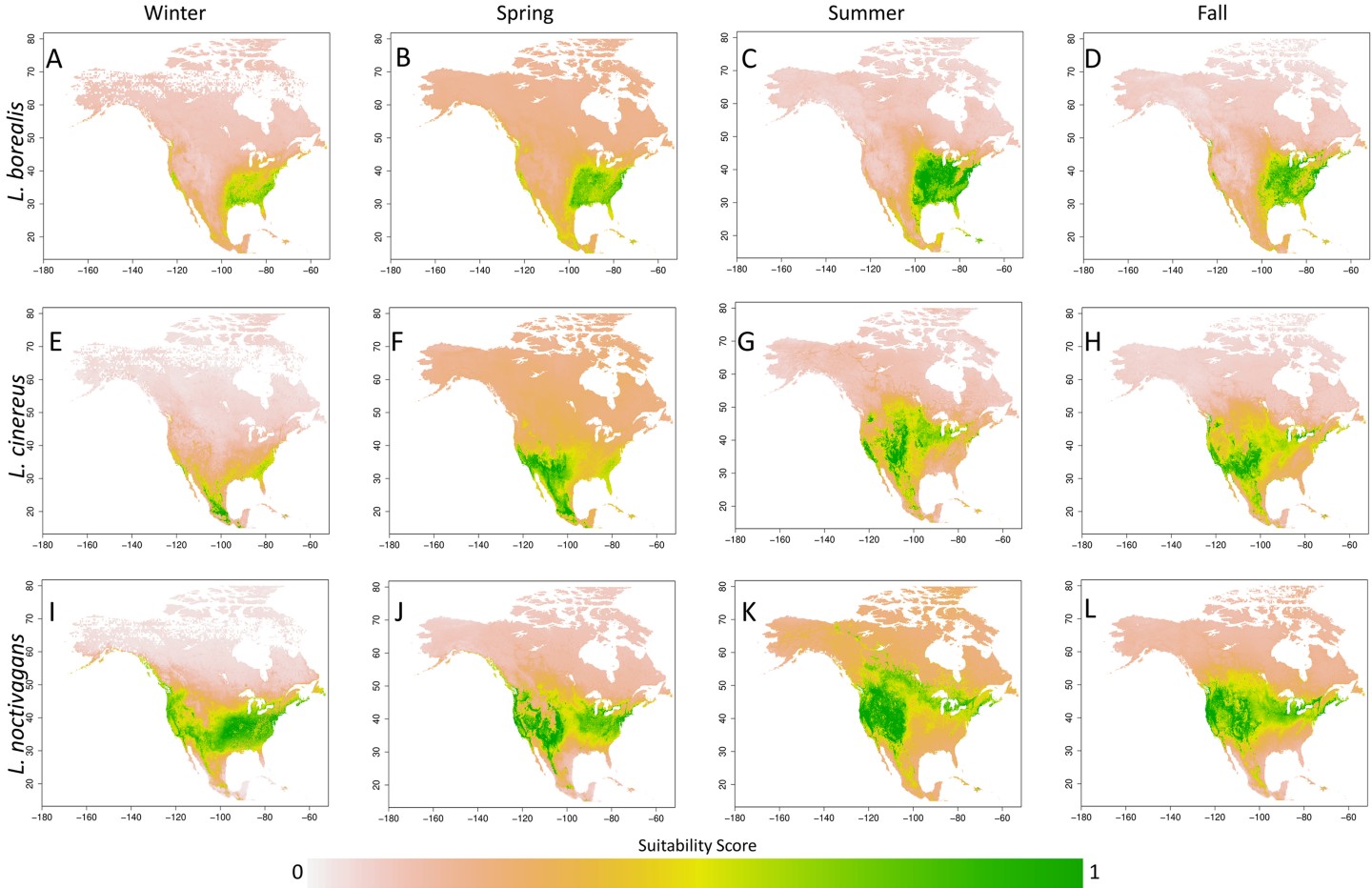

**Figure 3 Seasonal suitability for three Species of migratory bat species.** Seasonal SDMs for all three species (*L. borealis*, *L. cinereus*, and *L. noctivagans*). Colors identify either individual species or groups of species that occur in a given area. Outlier cells have been removed, and all rasters rescaled to range from 0 to 1. For more detailed figures for each species, see Figs. S2–S4. Each species is as follows (in order of Winter, Spring, Summer and Fall): (A)–(D) *L. borealis*, (E)–(H) *L. cinereus* and (I)–(L) *L. noctivagans*. Map outline generated using the 'maps' R package (*Becker et al., 2018*).

southern portions of their range during winter months with a northward movement during the summer into early fall. This is followed by a contraction again to the south. On the other hand, *L. noctivagans* exhibits a different pattern: while it has suitable habitat further north during the winter and expands northward it doesn't appear to do so to the same extent as the other two species. With these results being similar to expected, we can use them to predict the most likely migratory pathways.

## Potential migratory pathways

Using three methods (Circuitscape, LCP and 95th percentile), we find potential migratory pathways for each species that vary between the spring and fall seasons (Fig. 4; Circuitscape maps are shown in Fig. S5). In terms of spring migration patterns, *L. borealis* shows highest density of LCP along the Eastern coast of the U.S. and near the Mississippi River suggesting an avoidance of the Appalachian Mountains and using coasts and rivers as guidance during migration (Fig. 4). This pattern is also present in the 95th percentile maps.
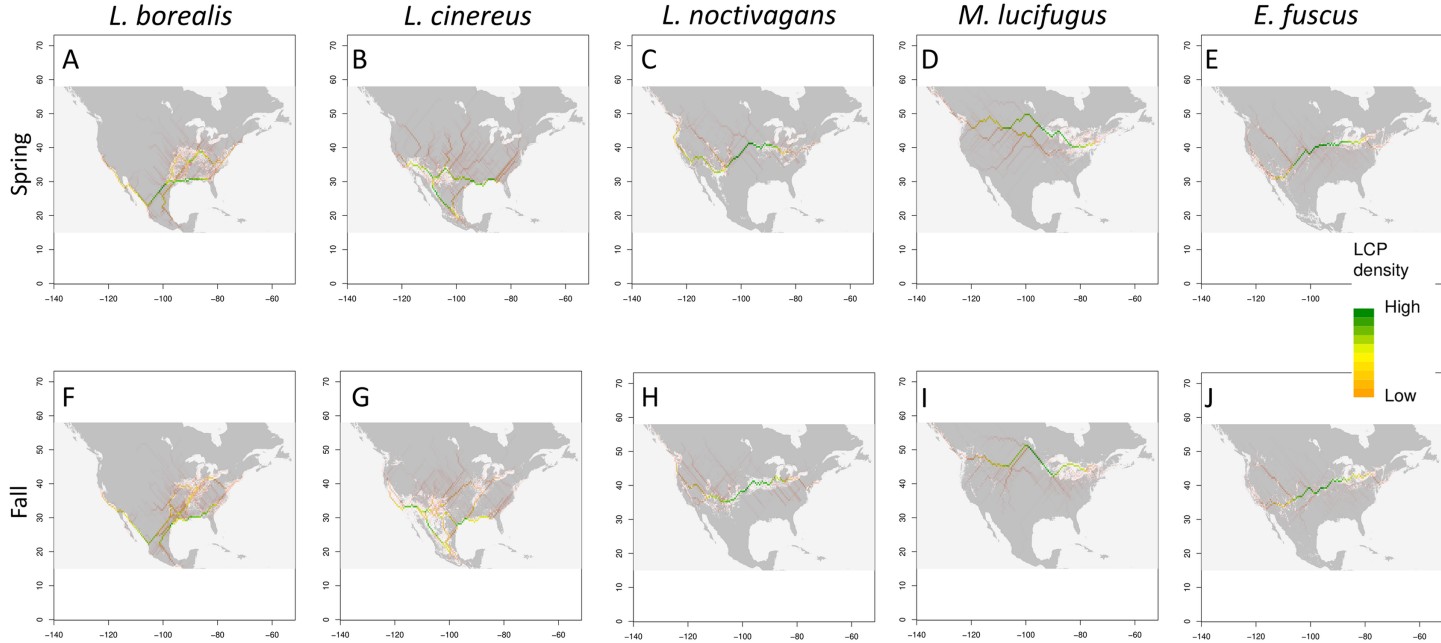

**Figure 4 Potential migratory pathways.** Migration pathways determined using two approaches: binary models determined from TSS weighted ensemble model using 95th percentile threshold determined for each species (shown in grey and white), and least-cost-path density (shown as color gradient) for each species (Spring and Fall respectively) (A) and (F) *L. borealis*, (B) and (G) *L. cinereus*, (C) and (H) *L. noctivagans*, (D) and (I) *M. lucifugus* and (E) and (J) *E. fuscus*. Map outline generated using the 'maps' R package (*Becker et al., 2018*).

For *L. cinereus*, higher LCP densities occur along Western Mexico into the Southern U.S., after which the higher probability pathways lie on either side of the Rocky Mountains pattern and along the Atlantic coast suggesting a lack of resolved pathway during this time period (Fig. 4). This is also supported by the 95th percentile map showing higher suitability scores being present in both these regions before the paths would extend further north. Finally, *L. noctivagans* shows two different patterns: LCP maps suggest movement from south to north in the Western U.S. along the Pacific coast and along the western edge of the Rocky Mountains (Fig. 4). In the Eastern U.S. there appears to be more of an east-west movement during which individuals would split off to move north or south, likely indicating a partial or incomplete migration in this species. For each of these species we see significant positive clustering in our pathways when using both Moran's I and Geary's C (Table 3). We can also observe that these potential pathways are significantly longer than straight line distance by hundreds of kilometers meaning these pathways would be biologically important, or in other words, that the most likely paths found here follow some biologic aspect of the area.

In terms of fall patterns, *L. borealis* shows two apparent migration paths: one along the East coast, and the other near the Mississippi River and into the Southern plains (Fig. 4). These paths are supported by the 95th percentile map, which shows suitable habitat in these areas at the same time of year. These two paths again indicate a potential following of coastline and rivers as linear guides during migration. *L. cinereus* shows evidence for multiple pathways (Fig. 4). Two possible pathways are present along the coasts of the

**Table 3 Spatial clustering of paths.**

| | Spring | | | | | | Fall | | | | | |
| --- | --- | --- | --- | --- | --- | --- | --- | --- | --- | --- | --- | --- |
| | Moran's I | | Geary's C | | Paired *t*-test | | Moran's I | | Geary's C | | Paired *t*-test | |
| | I | *p*-Value | C | *p*-Value | Mean increase | *p*-Value | I | *p*-Value | C | *p*-Value | Mean increase | *p*-Value |
| *L. noctivagans* | 0.39 | 0.01 | 0.60 | 0.01 | 880.40 | <0.001 | 0.34 | 0.01 | 0.67 | 0.01 | 767.16 | <0.001 |
| *L. borealis* | 0.45 | 0.01 | 0.54 | 0.01 | 348.33 | <0.001 | 0.42 | 0.01 | 0.56 | 0.01 | 325.39 | <0.001 |
| *L. cinereus* | 0.36 | 0.01 | 0.65 | 0.01 | 721.51 | <0.001 | 0.37 | 0.01 | 0.63 | 0.01 | 526.85 | <0.001 |

**Note:**
Moran's I and Geary's C to determine if clustering among potential migratory pathways is present. For Moran's I, values range between −1 and 1, with values above 1 indicating positive clustering. Geary's C values range between 0 and 2, with values below 1 indicating positive clustering. Results given for paired *t*-tests comparing Euclidean and Least-Cost distances.

Atlantic and Pacific, again indicating a possible following of coastlines during migration. While the Pacific is the clearer pathway of the two there is still a high density of lines along the Atlantic, which could be a minor pathway for those individuals navigating around the Appalachian Mountains. In addition, a pathway appears in our LCP map and is supported by the 95th percentile map along the Mississippi River. There is also evidence for movement through the Great Plains between the eastern slopes of the Rocky Mountains and the interior highlands near Missouri and Arkansas. Finally, *L. noctivagans* shows similar patterns for fall as those observed during spring migration periods (Fig. 4). We see a north-south pathway west of the Rocky Mountains, and east of those, a more east-west pathway is observed, with movements extending north or south, which again potentially indicates a partial or incomplete migration. With fall migration, we also observe positively clustered pathways that are significantly longer than Euclidean distance (Table 3).

The two bats that are not long-distance migrants show less variation in seasonal distribution as compared to the three migrant species discussed earlier (Fig. 4). In particular, both *E. fuscus* and *M. lucifugus* show a consistent east-west distribution pattern that does not change throughout the year. This supports the idea that changes in distributions of the migratory species likely reflect migratory behavior. Of interest is that the pathways determined by LCP for *E. fuscus* are similar to *L. noctivagans*, providing additional support that silver-haired bats undergo only a partial migration, that being some individual migrant while others overwinter in northern portions of the range. This seems possible as it has been documented silver-haired bats can overwinter at Northern latitudes (*Cryan, 2003*).

## DISCUSSION

Other studies have used SDMs and occurrence to model seasonal distributions of wide-ranging migratory bats including those studied here (e.g., *Cryan, 2003*; *Hayes, Cryan & Wunder, 2015*). This study extends this approach by using SDMs to predict migratory corridors, in this case the most likely path used. In essence, while others have used these data to delineate where bats are at a given time, we expand on this by attempting

to understand what is happening between these occurrences. Below we discuss limitations of our analyses and then expand on the conservation implications of our results.

## Analysis limitations

Using species occurrence data to generate species distribution models can be impacted by sampling biases present in the data (*Feng et al., 2019*). We attempted to minimize these biases by following guidelines described in *Feng et al. (2019)*. Specifically, we took steps to reduce oversampling of regions by subsampling our dataset to 1 point per 1° grid cell. Additionally, because occurrence records only represent presence points, and not true absences, we included models that require only presence data or can be adapted for use with presence only data. Despite these measures, it is possible biases remain in our models and so we stress that our models represent hypothetical species distributions and migratory pathways for any point in time. Another potential limitation with this approach is ability of bats to traverse unsuitable habitat. This could lead to some true pathways we are unable to predict as our models "prefer" suitable areas for inferring patterns of movement. However, the approaches used allow for some level of traversing areas of unsuitable habitat prior to using suitable stop-over sites (as noted above). As a result, we further stress that the pathways presented are only the most likely to be used and should be investigated further via other sources of data for inferring individual movements such as biomarkers and GPS tags.

## Migratory pathways

Our analyses identify potential migratory pathways across modelling approaches, although we observed some differences that likely result from features of the data that are given different weight by different methods (*McClure, Hansen & Inman, 2016*). For example, we were unable to identify clear pathways using Circuitscape despite using multiple transformations of our data (square root, log, natural log and cube-root transformations). The lack of identifiable paths using Circuitscape may indicate a true lack of clear migratory pathways yet still reflect the general patterns shown by the other methods. This is supported by the least cost path analysis, as while the figures present the most likely paths (Fig. 4), many other paths were evident (see Fig. S6). Diffusion or a wandering migration, across the landscape has been proposed for these species in previous studies (*Weller et al., 2016*; *McGuire, 2019*). As noted in *Baerwald et al. (2021)* for some species of migratory bats, more erratic 'wandering' movements in opposite direction of typical seasonal movements may be observed. While we find some evidence for this in our data, the Circuitscape maps identity areas of higher possible movement are also predicted by the LCP and 95th percentile threshold methods, providing support for specific proposed pathways. We emphasize that our results are not definitive delineations of a single migratory corridor followed by all individuals for the entire duration of migration. Rather, they identify the general paths followed during spring and fall migration while allowing for individual variation.

We note that specific features of our most likely migration pathways match patterns proposed by others. For example, in *L. cinereus* we find support movement along the

eastern slope of the Rocky Mountains. The same pattern was proposed by *Baerwald, Patterson & Barclay (2014)* as a likely route followed by spring and fall migrants of the same species to and from Alberta, Canada based on isotope data. In addition, consistent with our study, results from GPS tagging work suggest that *L. cinereus* uses the west of the Rocky Mountains during Autumn migration (*Weller et al., 2016*), although the sample size was limited. Finally, further work in *L. cinereus* using isotopes by provides strong support for predominantly north-south movement (especially in the Autumn) with some east-west movement *Cryan, Stricker & Wunder (2014)*. These patterns are also observed in our data.

In addition, while some differences between this study and those observed in *Cryan, Stricker & Wunder (2014)* such as apparent crossing of the Rocky Mountains, these could be in part due to how the pathways are presented. In *Cryan, Stricker & Wunder (2014)* they represent movement using straight lines, whereas in nature bats may follow less direct, non-linear paths which are better captured by our approach. Further differences can be observed for *L. cinereus* in that *Cryan, Stricker & Wunder (2014)* find less support for north-south movement during spring than we do. This difference could be an artifact of the use of isotope data which can lack precision, or it's possible our current understanding of spring migration is different than previously expected and tested here. *Weller et al. (2016)* also found some differences between our proposed pathways and their data. While their sample sizes were limited, they found some support for a more 'wandering' migration in this species (at least west of the Rocky Mountains).

Finally, in *L. borealis*, there have been numerous reports of individuals being captured offshore during migratory time periods (e.g., *Sjollema et al., 2014*; *Hatch et al., 2013*). These reports support our findings suggesting bats use linear features, such as coastlines, during migration. Lastly, *L. noctivagans* is regularly captured during winter months in the northern portions of their range (*Falxa, 2007*; *Barnhart & Gillam, 2017*) supporting the interpretation of this species as a partial migrant. However, as noted in *Baerwald, Patterson & Barclay (2014)*, some individuals are likely migratory and may follow some portion of the Rocky Mountains, a path not strongly supported in our data for *L. noctivagans*. The lack of clear pathways for *L. noctivagans* also support the 'wandering' migration proposed by *McGuire (2019)*. In summary, while some differences do exist between our results and others, the broader trends in terms of patterns of movements in these bats appear to be mostly consistent with previous studies.

The most likely pathways found here for bats match migratory patterns of many other species in North America including waterfowl and insects (e.g., *Lincoln, 1935*; *Westbrook et al., 2016*; *Tracy et al., 2019*). Of interest is the similarity to insect migrations which is consistent with an idea proposed by *Rydell et al. (2010)* that bat deaths at wind farms may be related to feeding on migratory insects near turbines. Bats may be tracking the migration of insects to determine their pathways and are feeding on them during migration leading to turbine mortality of bats (but see *Reimer, Baerwald & Barclay, 2018*).

Another possible explanation for the paths in the results are that bats use linear features, such as rivers, coastlines, and mountain ranges, as guides during migration (*Wang et al.,*

*2007*; *Ijäs et al., 2017*). For example, in *L. borealis* we observe apparent tracking of the Mississippi river and Atlantic Coast/eastern edge of Appalachian Mountains, while *L. cinereus* tracks the previous two mentioned and the Pacific Coast. One proposed rationale for the tracking of water bodies is that these features support a higher abundance of prey to feed upon during migration, allowing for more rapid travel (*Furmankiewicz & Kucharska, 2009*). We note, however, that other studies have failed to support this idea. For example, *Krauel, McGuire & Boyles (2018)* did not find evidence that another species of bat (*Myotis sodalis*) used rivers as a navigation guide during migration. Likewise, based on data from acoustic surveys, *Cortes & Gillam (2020)* did not find support for the use of the Missouri River as a migration guide for multiple species. These differences may be due to the geographic scale at which the studies were conducted compared to the results presented here. For example, *Cortes & Gillam (2020)* was conducted over a ~100 km of the Missouri River while our study focuses on much larger scales. It's possible that the 100 km portion of the river studied by *Cortes & Gillam (2020)* is not widely used as a linear feature for navigation but the use of rivers is common when looking more broadly. The last possibility for the apparent tracking of rivers and coasts is that increased tree cover also appears to follow these same features (i.e., near river = more trees; as observed in tree cover maps from *Hansen et al. (2013)*). While there are multiple hypotheses for the tracking of linear features, we are not able to distinguish between them, and it could even be some combination of them all. Further, the rationale for use of linear features likely vary among and between species of bats. As noted above, while all this may be true, we only present the most likely path for migration but that does not mean it is the only path. While individuals likely vary, the broad trends observed in our data can inform the conservation of these species.

Conservation for migratory bat species needs to be politically and geographically broad to be effective (*Fleming, 2019*). Conservation plans need to include protecting roost sites (during all stages of life), stop-over sites, and conserving foraging habitat around these sites (*Fleming, 2019*). Our results give direction as to where to look for stopover sites during migration, provide a starting point to identify areas where protecting habitat for migration is needed, and supply information as to where to best implement smart-curtailment mitigation methods during fall migration (*Hayes et al., 2019*).

In addition, there has been a recent focus on understanding the small-scale patterns of the movement that are embedded in larger migration patterns of these bats to develop effective conservation measures at small spatial scales (e.g., *Baerwald, Patterson & Barclay, 2014*). Our results contribute to this effort by providing specific hypothesis with which to direct future studies that focus on understanding small-scale aspects of the larger migration movements. For example, researchers could look for migration through regions highlighted here by using acoustic detectors or other methods along areas of the Mississippi River or in areas moving between coastal and nearby mountain ranges, similar to the work done by *Cortes & Gillam (2020)* along the Missouri River. This type of research could identify areas of high bat activity during migration periods where mitigation efforts could be focused to have the largest impact through reduction of mortality due to wind farms and other causes.

## CONCLUSIONS

Developing a better understanding about how these species move to and from summer habitat may be key in reducing the number of bats killed at wind farms. With bats making significant contributions to the economy of the United States through ecosystem services (*Boyles et al., 2011*) and provide valuable ecological services (*Ghanem & Voigt, 2012*) effective and practical measures are necessary to reduce the number of bat deaths annually at wind farms (*Frick et al., 2017*). By understanding migration, we can better mitigate and conserve species that are currently of concern in many states (e.g. *Ohio Division of Wildlife, 2015*). Our study provides SDMs that can be used as priors in conjunction with isotopic or other biomarker models for determining movement and more importantly, establish a proof of concept of how SDMs can be used to predict migratory pathways. We further provide more information on the movements of migratory bats, thereby informing researchers on where to focus our efforts towards the goal of reducing bat mortality due to wind farms.

## ACKNOWLEDGEMENTS

We like to thank the Carstens and Gibbs lab groups in the Department of EEOB at Ohio State for their help in proofreading and editing of this manuscript. We also thank Erin Hazelton and Jonathan Sorg, Ohio Division of Wildlife, for assistance with grant administration. This study is a contribution from the Ohio Biodiversity Conservation Partnership between Ohio State University and the Ohio Division of Wildlife.

### Funding

This work was supported by a grant (GRT00046616) from the Competitive State Wildlife Grants Program to Ohio State University and the University of Maryland Center for Environmental Science as jointly administered by the US Fish and Wildlife Service, the Ohio Division of Wildlife and the Maryland Division of Natural Resources. The funders had no role in study design, data collection and analysis, decision to publish, or preparation of the manuscript.

### Grant Disclosures

The following grant information was disclosed by the authors:
Competitive State Wildlife Grants Program: GRT00046616.
US Fish and Wildlife Service.
Ohio Division of Wildlife.
Maryland Division of Natural Resources.

### Competing Interests

The authors declare that they have no competing interests.

## Author Contributions

- Jamin G. Wieringa conceived and designed the experiments, performed the experiments, analyzed the data, prepared figures and/or tables, authored or reviewed drafts of the paper, and approved the final draft.
- Bryan C. Carstens conceived and designed the experiments, authored or reviewed drafts of the paper, and approved the final draft.
- H. Lisle Gibbs conceived and designed the experiments, authored or reviewed drafts of the paper, and approved the final draft.

## Data Availability

Data and code used are available at GitHub: https://github.com/jgwieringa/Seasonal_SDM.

Data is also available at Dryad:

Wieringa, Jamin (2021), Seasonal distributions and predicting migratory pathways for three species of bats, Dryad, Dataset, DOI 10.5061/dryad.2rbnzs7mz.

## Supplemental Information

Supplemental information for this article can be found online at http://dx.doi.org/10.7717/peerj.11177#supplemental-information.

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
