# Peer review of "Predicting migration routes for three species of migratory bats using species distribution models"

_PeerJ, doi:10.7717/peerj.11177_

## Round 0.1 · original submission · Major Revisions

We have now received two in-depth reviews for your manuscript. Both reviewers make a number of suggestions that I would encourage you to consider in your revision.

Most comments are around framing of the manuscript and/or focusing the writing.

I personally would also like to echo one of reviewer 1's comments:
"If you are not doing this already I would encourage you to post your data and code on a publically-accessible repository."

·

Basic reporting

This study uses quantitative modeling and occurrence data for several migratory bat species in North America to derive models and maps of species distributions and possible seasonal migration patterns for these species on a continental scale. The occurrence dataset was collected using Global Biodiversity Information Facility (GBIF) data. Once species distribution models (SDMs) were developed, several approaches to modeling seasonal migratory patterns (circuit theory, 95th percentile suitability, and least cost path analyses) were used to develop possible seasonal migration patterns on the North American continent. The manuscript is well-written with an introduction to the study and some natural history characteristics of these species.

My comments on the manuscript focus both on the quantitative methods and the level of circumspection in the analysis. I am not as well-read as I would like to be on the methods used to develop the migratory pattern models (circuit theory, least cost analysis, etc.), so I can’t offer much serious feedback regarding the modeling done using these methods. I do feel more comfortable in the species distribution modeling domain and so provide some suggestions for this piece of the analysis. Based on the details presented I suspect that the quantitative analysis is done reasonably well throughout. However, I don’t think the quantitative analysis is as well-explained as is could be, and the rationale for the approaches taken in the analysis are not always quite clear to me. I would encourage the authors to address the following methodological questions more thoroughly so other researchers and managers in this and other domains will more clearly understand the rationale and perhaps use this work as a framework to consider in their own areas of research. However, if the handling editor does not feel that these issues need to be addressed, I do not necessarily think they need to be addressed prior to publication, though some more detail (as described below) would likely help some readers, including myself.

Throughout the paper: consider defining “migratory corridor” early in the introduction so that readers know what you mean by this term, and then consider returning to this topic in detail in the discussion. I’m not convinced that the migratory tree bat species you analyze use migratory corridors per se. It is possible they do, but it seems to me that a reasonable hypothesis would be that they tend to more or less diffuse from winter to summer grounds and then back to winter grounds along broad geographic fronts. For example, hoary bats, being relatively solitary, may very well migrate from the southeastern United States where some populations spend winter into the interior of the continent where they spend the warmer summer months, and then back to the southeast in the fall. I think it would be reasonable to hypothesis that this general movement pattern may very well happen in a way that is more like gas particles moving from higher to lower pressure, or a liquid moving across a surface from higher to lower ground. If this is the case then describing the phenomena as a migratory corridor would probably not be an accurate description of the phenomena. Consider being clear about what you mean by the term corridor, and also consider explicitly stating that this is one possible hypothesis about migratory movement, but not the only possible pattern. This detail would add some length to the manuscript, but I believe it would be worth discussing in some detail.
Likewise, I believe your analysis assumes that these species follow gradients of habitat suitability and that the species are never really allowed to “decouple” from contact with the habitat suitability surfaces that shape the migratory patterns you are deriving. Some readers (myself included) will wonder whether these species, being long-distance migrants that often fly long distances at high altitudes above the ground, do indeed regularly decouple of the influence of these habitat suitability surfaces, and will thus wonder how this potential phenomenon is dealt with in your analysis. Some readers may also wonder if these species may regularly pass through and over lower quality habitat in order to move quickly between high quality winter and summer habitat. It would certainly seem reasonable to me to conclude that some if not all of the individuals of these species regularly decouple from the surface habitat suitability of a given pixel in your modeling template, and likewise regularly pass through lower quality habitat in transit. It is not clear to me how you have considered and resolved these questions, and I would encourage you to consider discussing these questions in some detail, especially in the discussion.

Thanks for the opportunity to take an early look at this manuscript. I enjoyed reading it and look forward to seeing it in print.

Experimental design

Generally I think the study design is very good and represents a novel and creative approach to analyzing seasonal migratory patterns of these species.

Line 113. Rationale for using only records from last 50 years. I don’t find this argument to be very convincing. For example, there are numerous records that could be used for specimens collected before about 1970. Many of these records can be definitively identified to species and many of these records can be reliably identified to their described occurrence location. See Cryan (2003) for a discussion of these details; Cryan compiled a comprehensive North American data set of the three species you analyze going back to the late 1800’s and in considered an authoritative sources.

Line 119. Consider adding roughly how many km 1 degree of latitude and longitude is equivalent to. I believe this is about 111 km or so. It seems to me that this would substantially thin the data, possibly in a way that causes the resulting models to not resolve relative habitat suitability very well. If implementing this course thinning procedure I would be particularly concerned that this would be the case in mountainous areas and in areas with substantial topographic relief. I would encourage you to discuss this in a bit more detail in this part of the methods. My opinion is that it’s likely that you have over-thinned the occurrence data.

Validity of the findings

My perspective is that the analysis is very likely done reasonably well, and thus the results are valid given the data and approaches used.

If you are not doing this already I would encourage you to post your data and code on a publically-accessible repository.

Consider comparing and contrasting your results and conclusions with those of others who have analyzed the migratory patterns of these species in North America and beyond. You very briefly mention the results of other workers but I think the manuscript would be much improved if you made sure to compare and contrast your results and conclusions with earlier workers in more detail. For example, the pathway analysis map in Figure 4 strikes me as a gross oversimplification of the seasonal migratory patterns that are occurring in reality. What have earlier workers said about these patters? Do your results confirm or contradict these ideas, and why?

Additional comments

Line 58 and throughout the paper, “latitudinal migrant”. My sense is that these bats aren’t really latitudinal migrants, per se, but rather tend to move from high quality winter to high quality summer habitat in ways that sometimes follows latitudinal gradients. So for some individuals and populations these patterns may follow latitudinal gradients, but not always. For example, some prior researchers have proposed that hoary bats move from areas of the North American continent that are closer to oceans and other large water bodies in winter to the interior of the continent in summer. This does not really follow latitudinal gradients. The same may be true along mountain fronts like the Rocky Mountains. Consider being clear about what you mean by latitudinal migrant and consider whether this is the term you want to use here and elsewhere in the paper.

Figs 2 & 3 could be sharper. I have a very hard time interpreting what’s being presented in these figures in the pdf. I think this is just a result of the lower resolutions of the figures in the pdf version offered for review, and it looks like the supplementary figures are reasonably sharp. But these are important figures in the overall argument of the paper and readers will want to see them clearly.

Fig 2. The color ramps for LABO and LACI extend from 0-0.6, but from 0-0.5 for LANO. Consider standardizing the color ramps so they are the same for all three maps. I would suggest scaling them all from 0-1.0. I also find the color ramp colors difficult to view. For example, summer and fall for LACI is very difficult for me to visualize and interpret.

Line 14. Consider removing “accurate” from the “describe accurate seasonally resolved distributions” phrase. This seems to imply that prior efforts are inaccurate.

Lines 17 & 16. Consider saying how many locations for each of the three species; and indicate which decades.

Line 93. Consider describing what the WorldClim data is. This will be helpful for readers who don’t know this lingo.

Methods and throughout. Consider checking to make sure that you have provided a good reasonably-recent citation associated with each SDM algorithm you use. You provide good citations for Maxent, but not for some of the other SDM approach. Some readers, including myself, will want to see high quality citations for these approaches in case they want to follow up and get more details about the approach.
Consider adding a figure to the manuscript that shows the locations of the occurrence locations used in the analysis, for example by showing a panel of maps, one for each species, that shows the locations of the points. Some readers, including myself, will want to see this. This would also allow readers to consider any sampling bias that may be associated with the locations, for example by looking for clusters of points and/or large gaps in occurrence records, etc.

Line 123. Consider saying approximately how many km are in 2.5 minutes of resolution (I believe about 4.5 km).

Lines 125-6. Consider adding the units used for each variable considered. For example, temperature (°C), winds

Reviewer 2 ·

Basic reporting

The authors present an interesting underlying pair of hypotheses (that seasonal migration habitats and pathways of certain bats can be predictively modeled from presence-only occurrence records), but in my judgement fall a bit short in a few areas of basic reporting. First, although things start out well and the writing is passably professional, things come a bit unwound in the latter half of the manuscript, where the writing begins to include contractions, grammatical errors, and generally begins to wander in terms of thematic consistency and focus. Hypotheses that are either explicitly stated or alluded to during the earlier parts of the manuscript are not consistently revisited in the context of the findings, whereas new ones appear in the latter parts of the discussion that surprised me. An example of where consistency checks could improve the professional feel of the writing include statements in the abstract saying there are "...many species of migratory bats...'' followed later by "...only 12 of 500 Vespertilionid bats undertake long-distance migration...". Overall, I think the depth and breadth of the background literature is somewhat shallow but probably sufficient, although several key concepts presented in the manuscript do not cite relevant primary literature (e.g., prior occurrence-record and stable-isotope evidence of longitudinal components to migration in these species). In general, I got the impression that the authors were under-describing the prior effort to hypothesize seasonal distributions of the hoary bat using these same SDM methods. I was not convinced the SDM elements of the study described here were unique enough to downplay those similarities (and spend text comparing results and addressing previously highlighted problems/biases inherent in that type of analysis) to the prior effort. Other things that rubbed me the wrong way while reading through this manuscript were things like use of the term 'latitudinal migratory bats' when that is not common use or necessarily accurate, as well as potential false dichotomy of migrating bats following linear features or not. The writing and logic style of the manuscript in its current form leave open opportunities for improvement.

Experimental design

If I read between the lines and ignore the somewhat opportunistically scattered hypotheses that surface later in the text, I think the real strength of this study is the novel use of what I interpreted as diffusion models to try and improve understanding of where migrating bats might concentrate in landscapes between seasonal occurrences. This is a really neat idea and it makes sense that given the best information at hand (temporally variable and spatially explicit occurrence records), the next logical step in trying to infer the seasonal movements of these elusive migratory animals is to look at transitions between known and hypothesized known locations. The earlier work to model seasonal species distributions of these tree bats using museum occurrence records, then stable isotopes, and then SDM methods all took understanding from a single, static species distribution map to what is essentially a seasonal and now monthly time series of species distribution maps. Extending the analogy, the prior work focused on the data points along the time series, and now the current study begins addressing what might happen between those points. In my opinion, this manuscript could be much improved by focusing less on the tangential and tenuous underlying justifications of the work (e.g., that yellow warblers are observed more often because they are active during the day...a rabbit hole I will avoid here), claiming less novelty with the SDM ensemble modeling, and honing the manuscript to support the assumptions, methodological details, and findings of the transition model elements. You can tell from my loose use of terms that I'm not sure whether to refer to the intra-month modeling as diffusion, transition, flow, or something else, but further making that part pop out with terminology consistent with the scientific literature would really help. Put another way, future scientists wanting to replicate this work can go to other studies to get details of the SDM methods used, but will not have other leads in the bat literature for repeating the diffusion modeling to study changes in density of migrants across landscapes at various spatial and temporal scales.

Validity of the findings

The authors' use of diffusion modeling methods to generate spatially and temporally explicit hypotheses of variation in density of migrating tree bats was the strongest element of this work that deserves further fleshing out and exploration. It was good to see additional SDM modeling using methods very similar to a prior effort, but the claimed increase in resolution brought with improving the sampling period (from seasons to months) and spatial resolution of environmental covariates (down to 2.5km grid cells for climate info) was not presented with comparatively convincing evidence to support the claim of implied improvement. That seems incidental to the overall impact if focus were shifted to the month-to-month transition analysis. More information exists in the literature presenting alternative migration patterns to the simple latitudinal model implied in this manuscript as the prevalent standout (e.g., multiple stable isotope studies of all three species of tree bat)---inclusion of those alternatives in the context of the diffusion model findings could also provide new and unique context.

Additional comments

Despite my reservations about the way the information is currently presented. The underlying findings of the diffusion modeling to explore and model density of migrants are exciting and promising.

---

## Round 0.2 · accepted · Accept

Thank you very much for addressing the reviewer comments.

One of the previous reviewers and I went over your revisions and we are both happy with how you addressed them.

Thanks!

·

Basic reporting

The reporting in the revised manuscript is clear and thorough.

Experimental design

The experimental design is appropriate for this analysis and a creative approach to the problem.

Validity of the findings

I believe the results are valid given the data and approaches used.

Additional comments

Nice work on this paper and keep up the creative modeling and analysis.